# Improving Vascularization of Biomaterials for Skin and Bone Regeneration by Surface Modification: A Narrative Review on Experimental Research

**DOI:** 10.3390/bioengineering9070298

**Published:** 2022-07-04

**Authors:** Heiko Sorg, Daniel J. Tilkorn, Jörg Hauser, Andrej Ring

**Affiliations:** 1Department of Plastic and Reconstructive Surgery, Marien Hospital Witten, Marienplatz 2, 58452 Witten, Germany; heiko.sorg@uni-wh.de; 2Department of Health, University of Witten/Herdecke, Alfred-Herrhausen-Str. 50, 58455 Witten, Germany; 3Department of Plastic, Reconstructive and Aesthetic Surgery, Hand Surgery, Alfried Krupp Krankenhaus, Hellweg 100, 45276 Essen, Germany; daniel.tilkorn@krupp-krankenhaus.de (D.J.T.); joerg.hauser@krupp-krankenhaus.de (J.H.); 4Department of Plastic, Reconstructive and Aesthetic Surgery, Hand Surgery, St. Rochus Hospital Castrop-Rauxel, Katholische St. Lukas Gesellschaft, Glückaufstraße 10, 44575 Castrop-Rauxel, Germany

**Keywords:** vascularization, lactocapromer, polylactide, polymers, bone substitutes

## Abstract

Artificial tissue substitutes are of great interest for the reconstruction of destroyed and non-functional skin or bone tissue due to its scarcity. Biomaterials used as scaffolds for tissue regeneration are non-vascularized synthetic tissues and often based on polymers, which need ingrowth of new blood vessels to ensure nutrition and metabolism. This review summarizes previous approaches and highlights advances in vascularization strategies after implantation of surface-modified biomaterials for skin and bone tissue regeneration. The efficient integration of biomaterial, bioactive coating with endogenous degradable matrix proteins, physiochemical modifications, or surface geometry changes represents promising approaches. The results show that the induction of angiogenesis in the implant site as well as the vascularization of biomaterials can be influenced by specific surface modifications. The neovascularization of a biomaterial can be supported by the application of pro-angiogenic substances as well as by biomimetic surface coatings and physical or chemical surface activations. Furthermore, it was confirmed that the geometric properties of the three-dimensional biomaterial matrix play a central role, as they guide or even enable the ingrowth of blood vessels into a biomaterial.

## 1. Introduction

Thermal and mechanical lesions of the soft tissues because of burns, decollement and crush injuries, open comminuted fractures, and soft tissue and bone infections such as phlegmons, necrotizing fasciitis, and osteomyelitis can cause extensive cross-layer tissue destruction. Here, reconstructive surgery is always limited by the restricted availability of donor tissue to cover the defect of the skin or soft tissue damage. In addition, the not negligible morbidity of the donor regions must be accepted in autologous tissue reconstruction. Artificial tissue substitutes, which can be produced within the framework of modern tissue engineering, are therefore of great interest for reconstructive surgery. The core idea of tissue engineering is to use biomaterials, cells, and bioactive molecules to help an injured tissue or organ regenerate. The use of artificial tissue substitutes in the treatment of critical tissue defects and substance loss also offers the possibility of reducing or even eliminating morbidity in or of donor regions. Over the past few decades, the field of tissue engineering has evolved greatly, and various strategies have been developed to accelerate and enhance tissue regeneration. It should be noted, however, that developments to date in the clinical application of tissue substitutes, except for endoprosthetics, have been rather disappointing. Even in experimental approaches, biocompatibility and their insufficient function in the recipient tissue is a major challenge [1,2].

Therefore, the development of artificial tissue substitutes for complex tissues, such as skin and bone, continues to require intensive research [3,4,5,6].

In the regeneration of the skin, there is always the question of the type of wound (acute, chronic, infected) and wound depth, in the sense of how much of the complex skin structure is injured and must be replaced. As an example, the treatment of burn wounds clearly shows that in patients with 50% involvement of the body surface, only 50% of undamaged skin is available for the harvesting of split skin grafts. This is a wound area that would cover 100% of the body. Therefore, especially in burn wounds, a fast-healing skin substitute plays an essential role and can ensure survival. Figure 1 shows skin substitute materials used to date.

In bone regeneration, it is critical size defects or osteomyelitic changes that play a role. Furthermore, the significantly slower regeneration of bone must be taken into account. For the treatment of bony defects, autogenous bone transplantation is still considered the gold standard. However, the number of possible bone substitute materials has increased significantly in recent years. The bone substitute materials are differentiated and classified according to their origin (Table 1).

Biomaterials used as scaffolds for tissue regeneration are non-vascularized synthetic tissues and often based on polymers. The ingrowth of new blood vessels into a tissue substitute is necessary to ensure the supply of nutrients and oxygen as well as the removal of metabolic products. For an artificial tissue construct to survive, the ischemia time after transplantation must be as short as possible [2,10,11]. This review summarizes previous approaches and their limitations, and highlights advances in vascularization strategies after implantation of surface-modified biomaterials for skin and bone tissue regeneration.

## 2. Methods of Bioactive Modification in Skin and Bone Regeneration

### 2.1. Challenges in Skin and Bone Tissue Regeneration with Synthetic Biopolymers

Biomaterials made of biopolymers are widely used for tissue engineering. This is due to their microstructural interconnectivity and inherent bioactivity, mimicking the natural extracellular matrix (ECM) and supporting cellular functions. By blending natural or natural with synthetic biopolymers and further physicochemical crosslinking treatments, the microstructure, mechanical properties, biostability, and cell activity of natural biomaterials can be controlled to provide the required mechanical strength, degradation rate, and ECM-mimicking microenvironment to support cell activity. An essential requirement for using degradable synthetic biopolymers as composite skin substitutes is the ability to establish new vascular tissue. Using a biodegradable synthetic polyethylene glycol terephthalate/polybutylene terephthalate (PEGT/PBT)-block copolymer matrix, we have previously demonstrated through histological studies that this biopolymer is only able to achieve complete vascularization after three weeks of implantation, even after optimization of its structural properties (i.e., larger pore sizes 75–300 µm) [12]. For clinical application, this means that the delayed vascularization of this material used as dermis substitute allows an only two-stage transplantation of split skin or keratinocytes at the earliest after a three-week interval. From a clinical point of view, this period appears too long for the PEGT/PBT dermis substitute material to be a useful tool in the treatment of severe burn injuries. A similar problem arises with bone graft substitutes. Sufficient vascular supply of a grafted bone substitute material is a basic requirement for stable integration of the biomaterial and induction of bone regeneration. However, analogous to dermis substitutes, and lack of or insufficient vascularization is also a major problem in the clinical setting.

### 2.2. Bioactive Modification for the Improvement of Skin Wound Healing

The rapid increase in understanding of matrix biology has opened the possibility of using the natural ECM as a model for developing new scaffolds and improving their effectivity through further modification. To study this, an antimicrobial peptide (AMP) was instilled in a biodegradable PEGT/PBT-matrix, to investigate whether this bioactive modification might influence the neovascularization [13,14,15,16,17,18,19]. For this purpose, the AMP cathelicidin LL-37 was used, which has already been very well studied for several years and has a pleiotropic activity profile [20]. Aside from its broad spectrum of antimicrobial activity, LL-37 also possesses antifungal [21,22] and antiviral properties [23,24]. Interestingly, it is also involved in the inhibition of biofilm formation, a very important function for skin wound healing as well [25,26]. The technique of intravital fluorescence microscopy was used to analyze the microvascular changes after implantation of the scaffold into the dorsal skinfold chamber and instillation of LL-37 over a 24-day period. This study could demonstrate a pro-angiogenic effect for cathelicidin LL-37. After instillation of LL 37 into the biopolymer matrix, acceleration of neovascularization was observed. These data indicate that cathelicidin LL 37 not only possesses multifunctional properties in regulating the immune response, but also plays an important role in the bioactive modification of biopolymers to enhance angiogenesis [15]. In further work, a plasmid encoding LL-37 was introduced into wounds by skin-targeted electroporation [27]. This technique showed that LL-37 was significantly integrated in the re-epithelialization of diabetic and non-diabetic wounds, and this was associated with an improvement of angiogenesis by induction of vascular endothelial growth factor-a (VEGF-a) [27]. Aside from matrix biomodification of scaffolds, polymers themselves could be modified. Chereddy’s group therefore used polylactic co-glycolic acid (PLGA), which was shown to support wound healing as well as durable drug release [19,28]. Additionally, PLGA revealed distinct positive effects in angiogenesis. Therefore, PLGA has also become the focus for ongoing research by modification with the AMP LL-37. The PLGA/LL-37 construct was administered to full thickness excisional rodent wounds and showed significant acceleration of wound healing, as given by advanced granulation tissue formation, higher re-epithelialized tissue, and improved angiogenesis (up-regulated VEGFa expression) [19]. In a recent study, LL-37-peptide was also encapsulated into chitosan hydrogel, due to its low stability in the wound environment [29]. This approach demonstrated that packaging LL-37 in chitosan hydrogel could significantly improve wound healing of experimental rodent pressure ulcers. The increase of newly formed capillaries as well as the increase in epithelial thickness were responsible for the promising application and demonstrated the high biocompatibility of this LL-37 formulation [29].

In summary, it can be said that a bioactive modification of, for example, AMPs can significantly promote tissue engineering for skin wound healing. The use of LL-37 in particular shows that this is not only limited to the function of wound closure but is also important for essential aspects of neovascularization.

### 2.3. Bioactive Modification for the Improvement of Bone Regeneration

Due to its broad spectrum of activity, AMP cathelicidin LL-37 also had beneficial properties in bone regeneration. Zhang’s group was able to show that blood-derived cathelicidin LL-37-differentiated monocytes form bone-like structures analogous to endochondral bone formation, both in vitro and in vivo [30]. In an ongoing study, this new cell line, called as monoosteophils, could be clearly differentiated by their gene expression profile from monocytes, macrophages, and osteoclasts. Monoosteophils clearly upregulated integrin α3 and down-regulated CD14 and CD16 [31]. As severe periodontitis may be related to LL-37 deficiency, Supanchart et al. investigated the role of LL-37 in osteoclastogenesis [32]. The findings of this study revealed that LL-37 was able to block osteoclastogenesis by inhibiting the calcineurin activity [32]. Bioactive modification with LL-37 on titanium implants further showed that this AMP can facilitate new bone formation via the recruitment of mesenchymal stem cells [33].

Other active peptides were excellently reviewed in the article by Wang and colleagues, categorizing the peptides in ECM-derived, bone morphogenetic proteins-derived and peptides [34].

### 2.4. Biomimetic Surface Coating for the Improvement of Skin Regeneration

The imitation of the porous ECM-topography is beneficial for the effective regeneration of damaged tissue. This can be achieved on the one hand by materials per se prone to porosity, e.g., po-lyethylene [35], but also by their manufacturing process, e.g., electrospinning, or form of delivery, e.g., sponge-, fibrous-, or gel-type scaffolds [36]. Although many of these porous biomaterials support the regeneration process, an additional improvement of the mode of action can be achieved by a biomimetic surface coating to stimulate new blood vessel formation and optimize the vascularization of the biomaterials.

In this context, CaCuSi_4_O_10_-containing compounds were shown to have pro-angiogenic effects [37]. Novel Chinese sesame stick poly(ε-caprolactone) and poly(D, L-lactic acid) scaffolds, whose surface was coated with CaCuSi_4_O_10_, demonstrated improved re-epithelialization as a result of increased angiogenesis [38]. This was especially demonstrated by a markedly increased blood vessels count and a higher density of CD31-positive vascularity in histology tissue specimen [38]. However, the actual mechanism of action was thought to be the release of Cu^2+^ and SiO_4_^4^-ions that upregulate VEGF expression [39,40,41].

Growth factor release is also an intriguing area of research to improve the vascularization of biomaterials. Platelet lysates represent a natural source for a large pool of growth factors. In a special preparation of a bioactive construct consisting of poly-l-lysine (PLL) and hyaluronic acid (HA) on free-standing polyelectrolyte multilayer films (PEMs), platelet lysates have been cross linked [42]. These films were studied in vitro and in vivo. Results revealed that the platelet lysate loaded films significantly supported primary fibroblast and primary human umbilical vein endothelial cell (HUVEC) proliferation, adhesion, migration, and angiogenic tube formation (HUVEC only) [42]. In a rat model of full excisional skin wounds, the animals treated with loaded films already displayed a neo-epidermis-like epithelium by day 7, while controls did not show any regeneration. By the end of the experiment, the loaded film treated animals showed a well-organized, multilayered skin epithelium, with an increased blood perfusion in highest platelet lysate group only. This was associated with the fact that this group also showed increased neovascularization by immunohistochemistry, as given by higher CD31-staining, VEGF-expression, and hypoxia inducible Factor 1α-levels [42].

### 2.5. Biomimetic Surface Coating for the Improvement of Bone Regeneration

The field of biomimetic surface modification to enhance bone regeneration is vast and quite well studied. Our own group investigated the effect of biomimetic surface modifications on vascularization using calcium phosphate- and collagen-I-coated PEGT/PBT-copolymer matrices [43]. Neovascularization of the matrices was analyzed by intravital fluorescence microscopy in the mouse dorsal skinfold chamber over a 16-day period. The intravital microscopic observations showed that the intensity of new blood vessel formation in the peripheral zone of the implants as well as the speed and intensity of the extent of microvascular penetration in the center of the porous matrices differed significantly depending on the surface coating. Although type I-collagen is a cell-adhesive matrix protein that plays an essential role in angiogenesis, surface coating of the copolymer matrices with type I-collagen did not show a significant effect on the acceleration of neovascularization in our study. During further investigations, we found that surface modification of the PEGT/PBT-copolymer matrix with calcium phosphate coating, which ultimately meant mineralization of the porous copolymer matrix to increase its potential osteoconductive properties, resulted in accelerated neovascularization of the modified matrix. In contrast, new vessel formation was comparatively reduced in the untreated matrices and in the collagen-coated matrices [43]. Therefore, calcium-mediated signal transduction plays an important role in angiogenesis [43,44,45,46]. The extent to which a change in local calcium concentration due to calcium phosphate coating shows an effect on neovascularization remains unclear and is the subject of ongoing investigation.

Aside from calcium, copper (Cu) might play a role in bone tissue vascularization [47]. Via magnetron sputtering, Cu can be applied to hydrogenated amorphous carbon films, which are then used as bone implants [48]. This method can also be applied to determine the roughness of the layers. Films with different roughness were subsequently subjected to extensive testing [48]. In the in vitro experiments with human bone marrow derived stem cells (hBMSCs), the films with high roughness showed significantly higher gene expression of VEGF after 21 days. Subcutaneous implantation of the Cu-coated films in rabbits showed increased regeneration of blood vessels in the vicinity of the implants compared with uncoated control implants. Interestingly, the higher the roughness of the films, the denser the neovascularization process. This shows that biomimetic Cu coating strongly improves the expression of the vascularization response in bone regeneration [48].

In bone regeneration, the process of osseointegration is of certain importance for the implant healing rate. For this, an Australasian research group developed a biomimetic implant coating by mimicking structures of the natural ECM of the host bone and the bone healing hematoma with titania fiber like nets on titanium implants [49]. Using in vitro assays, they demonstrated that these implants could facilitate osteogenic and angiogenic differentiation of bone marrow stromal and endothelial cells, while in vivo completely underscored these results. The complete mechanism of its biological effect could not clearly be clarified, however, authors hypothesized that modulation of the biomimetic structure might have a positive effect on the spatiotemporal regulation involved during osseointegration [49].

### 2.6. Physicochemical Surface Changes by Cold Plasma in Skin and Bone Regeneration

Physical plasma is the fourth state of matter after solid, liquid, and gas and has become increasingly important in medical research in recent years. Since plasma can be generated at room temperature, it can also be applied to sensitive tissue or materials. What makes plasma so extraordinary and at the same time so complicated are its various, diverse components and their effects on cells and fabrics. Cold plasma is always generated by a flow of energy between two electrodes. The term direct or indirect plasma refers to the way the plasma is directed to the surface and whether a neutral carrier gas or ambient air is used. The most fundamentally researched area in plasma medicine is sterilization. Even short plasma application of a few seconds significantly reduces bacterial density [50,51]. It is assumed that the destruction of bacterial DNA by UV radiation and the uptake of reactive species and the associated modification and destruction of proteins cause the toxic effects [50,52]. Even biofilm-forming and multiresistant germs that were previously difficult to treat could be significantly reduced by plasma treatment [53,54].

The important role of reactive oxygen (ROS) and nitrogen species (RNOS), which influence cell metabolism, was repeatedly pointed out. Many studies have shown that plasma effects are reduced or no longer detectable by artificial inhibition of ROS and RNOS. The reactive species can trigger the described effects via direct interaction with DNA. Arndt et al. were able to show that after cold plasma treatment of fibroblasts with an indirect plasma source, there was increased gene expression of key genes involved in wound healing [55,56,57]. These include genes that control the release of cytokines as well as the production of growth factors [55,56,57].

Neoangiogenesis is promoted by oxygen deprivation, cytokines, and growth factors, but also by ROS and RNOS [58]. Arjunan et al. showed that plasma treatment leads to increased proliferation and formation of in vitro endothelial cells. This seems to be caused by FGF-2, which is released by plasma-generated ROS [59]. Cytokines relevant for angiogenesis were shown to be released by fibroblasts and keratinocytes, and neoangiogenesis was increased in plasma-treated wounds of mice [57].

Ongoing studies were conducted to determine whether physical surface activation of biopolymer matrices, developed for use as synthetic dermis substitutes, leads to an acceleration of blood vessel growth into the matrices by altering surface energy due to plasma electrical discharge [60]. Analyses were performed by intravital microscopy after implantation of the biomaterials into the dorsal skinfold chamber of the mouse. In previous studies, accelerated sprouting of blood vessels in plasma-treated PEGT/PBT-copolymer matrices was observed [43]. In subsequent observations, acceleration of new vessel formation as well as significantly increased blood vessel density at days 5 and 10 after implantation, compared to untreated matrices, were observed at the implant margin of plasma-treated collagen/elastin matrices. A possible explanation for these observations is the plasma-induced increase in surface energy. The interaction of the material surface during a plasma discharge with released high-energy particles, such as ions and electrons, can lead to the formation of free bonding sites on the material surface, causing an increase in surface energy [61,62,63]. The results of the current work show that the plasma-induced increase in surface energy was able to lead to an increased vascularization [60].

Consequently, Griffin’s group was able to conduct a lot of studies on tissue integration and vessel formation of implanted synthetic biomaterials after plasma modification of the surface and topography changes [64,65,66].

Firstly, this group used plasma for two processes. On the one hand, oxygen plasma was used for surface activation; and on the other hand, plasma was used to functionalize the surface of the nanocomposite scaffolds with NH2 and COOH groups [64]. The physiochemical characterization of the scaffolds showed significant changes in their surface properties, as given by surface roughness and chemistry. In vivo experiments revealed no fibrous capsule around the implant as a sign of no adverse foreign body reaction. This was in comparison with significantly increased endothelial migration and neovascularization given by a greater extent of CD31 positivity [64].

Plasma surface modification was performed by subjecting polyurethane polymers (PUP) to different gases and exposure times, resulting in different roughness in their 2018 study [65]. The use of argon for plasma modification resulted in the overall best effects. Argon modified matrices showed moderate interfacial properties (i.e., chemistry, topography, and wettability), the highest protein adsorption, a significant upregulation of adhesion-related proteins and ECM marker genes, but above all, the highest tissue integration and angiogenesis and the lowest capsule formation in the in vivo experiments after subcutaneous implantation [65].

In a third study by Griffin et al., argon plasma was used to improve tissue-engineered scaffolds by adipose-derived stem cells (ADSCs) [66]. The highest level of tissue integration and neovascularization could be seen in argon pretreated PUPs with ADSCs [66]. The additional treatment with platelet rich plasma had no trigger effect. Argon can thus create a hydrophilic surface by depositing hydroxyl groups on the implant surface [64]. Argon modification therefore produced optimal topographic and surface chemical changes that enabled improved ADSC adhesion and consequently supported tissue integration and angiogenesis in vivo.

For the biological integration of implants, a strong bond with the surrounding tissue is necessary. The development of surface coatings, especially when using proteins, is difficult to realize due to the hydrophobic surface properties of most materials. Plasma-mediated surface coating with cold plasma and a matrix protein could be a possible solution. In vitro studies demonstrated for plasma-mediated collagen coating of titanium alloys, that this method leads to improved growth behavior of osteoblastic cells [67]. Based on the data of these observations, it was investigated whether the plasma-assisted collagen coating of a metal implant might also have an influence on the integration of the implant into the soft tissue [68]. The intensity of vascularization of the implant site served as the basis for assessing improved soft tissue integration. In vivo analyses were performed on plasma collagen-I-coated titanium implants after their implantation into the mouse dorsal skinfold chamber. Analysis of blood vessel density showed that plasma collagen-I-coated titanium implants had significantly increased new blood vessel formation in the marginal zone to the surrounding soft tissue compared with uncoated implants. Based on our observations, it can be concluded that plasma collagen-I-coating of titanium implants leads to an increase in blood vessel density in the soft tissue close to the implant and thus to an improvement of implant integration [68].

The increased and accelerated blood vessel formation as an expression of increased biocompatibility can be explained by improved adherence of mediators and proteins released from the wound environment to the plasma-activated surface of the biomaterial.

### 2.7. Modification by Changing the Surface Geometry

The biomaterials that have been studied in large numbers have in common that they often have insufficient geometric surface parameters to inhibit blood vessel ingrowth [69,70,71]. In the circulatory system, blood vessels represent a unique anatomical structure, with their innate physiology forming the basis for mechanical support against blood pressure, vasoactive response, and cell functionality. Recently, biomaterials engineering has explored the design of anisotropic surface geometries, which analyzed direct biological effects on the replication of vascular tissue architecture. Anisotropy here refers to the ability of certain cells to exhibit different growth directions under the same conditions. A vascular scaffold, therefore, requires to be a biomaterial which is biodegradable/resorbable, biocompatible, and hemocompatible. It must also display with robust mechanical properties and a large porosity enough for cell infiltration, cell–cell communication, and mass transportation. In further experiments, the vascularization potential of a synthetic dermis substitute material, a highly porous, biodegradable 3-D matrix of lactocapromer terpolymer (porosity of 85–93%; pore sizes of 50–400 μm; Figure 2b,c), based on polymers polylactide and polycaprolactone, was investigated [72]. The PEGT/PBT-copolymer was used as a control and displayed pore sizes of 250–300 µm, provided appropriate hydrophilicity to the copolymer and allowed high molecular permeability [12]. The results demonstrated a clear difference between the matrices studied, in terms of the time at which the first signs of new vessel formation occurred and the speed and intensity of neovascularization. For the lactocapromer terpolymer 3D matrix, there was an earlier onset and accelerated formation of new vessels at the edge of the implant compared to PEGT/PBT-copolymer matrices. The density of matrix-infiltrating and perfused blood vessels was 24% higher in the lactocapromer terpolymer matrix than in the PEGT/PBT-group on day 10 of the study. Analysis of the biomaterials by scanning electron microscopy confirmed the intravital microscopy observations in the quantification of the newly formed blood vessels within the matrix (Figure 2d). The results of this work clearly demonstrate the essential role of the matrix surface geometry of a biomaterial in the formation and sprouting of new blood vessels [72].

The restriction of tissue perfusion is not only determined by the extent of the injury. It can also be affected by inadequate wound management. In the following, the influence of a lactocapromer terpolymer matrix (porosity 70–80%; pore size 5–50 μm) as a temporary synthetic skin substitute material was investigated on local microcirculation and blood vessel formation in a dorsal skinfold chamber wound model [5]. As a result of intravital microscopic analyses over a 10-day period, we found a significant increase in new blood vessel formation, particularly in the marginal areas of the implanted matrices. This was accompanied by an increase in functional blood vessel density because of the formation of a new microvascular network. No impairment of microperfusion of the surrounding wound tissue was observed. On the other hand, despite the relatively low thickness layer of the here used matrices (less than 200 μm), no increasing vascular infiltration into the matrix was observed. One explanation for this observation was the structural differences between the temporary skin substitute and the previously studied dermis substitute [72], although both biomaterials have identical chemical composition. The results of both studies [5,72] showed that the intensity of sprouting of newly formed blood vessels in synthetic polymer matrices correlated closely with their surface structure.

Despite the increasing optimization of the mechanical properties and biological characteristics of bone graft substitutes based on calcium phosphate compounds, severe postoperative inflammatory reactions after their implantation were still reported [73,74,75,76,77], leading to their failure. In the following, resorbable calcium phosphate bone cements and ceramic calcium phosphate granules were analyzed, regarding an inflammatory soft tissue reaction after implantation [78]. The bone substitute materials used are all characterized by different composition, porosity, as well as pressure resistance (Table 2). Changes in functional blood vessel density, capillary stability, and leukocyte-endothelial interaction were analyzed by intravital microscopy as criteria for an inflammatory reaction of the soft tissue. The data showed differences between the groups of bone substitutes studied in terms of the mean increase in blood vessel density. It was found that the differences in the intensity of blood vessel formation were mainly related to the relatively large variations in the structural or geometric surface properties of the different used bone substitute materials. Increased functional blood vessel density was found for the resorbable bone cements and granules with high porosity which, at the same time, have a high proportion of macropores with larger diameter and thus higher pore connectivity [78]. The results on the microvascular response to bone graft substitutes confirmed comparable observations in the analysis of synthetic skin polymer matrices regarding the importance of the surface geometry of the materials used [5,72].

Adequate vascular supply of a grafted bone substitute is a basic requirement for stable integration of the biomaterial and for induction of bone tissue regeneration. The use of allogeneic bone in addition to synthetic bone cements might be an alternative in situations where there is no possibility of autologous bone grafting or larger amounts of material are required. In the following, the change in surface properties was investigated by treating decellularized cancellous bone substance with cold low-pressure plasma [79]. Furthermore, surface modifications were analyzed in order of increases in the vascular integration of bone chips in vivo [79]. Cancellous bone chips were analyzed by X-ray photoelectron spectroscopy (XPS analysis) before and after treatment in a plasma reactor. Microvascular and inflammatory responses in the implantation area of the cancellous bone chips as well as their vascularization behavior were analyzed over a period of 10 days. The surface analyses performed showed that surface modification because of activation by cold low-pressure plasma leads to a change in the chemical elemental composition of the cancellous bone surface. In addition to the surface chemical composition, a change in the chemical bonding state of carbon in terms of an increased amount of oxygen-bonded carbon was observed on the surface of the plasma-treated bone chips compared with controls. Data analysis of the intravital microscopic observations revealed that surface activation of the allogenic bone resulted in increased formation of new blood vessels, and thus improved integration of the bone chips into the soft tissue. In conclusion, the microvascular density at the implant site and the microvascular infiltration of the decellularized cancellous bone substance were intensified by plasma surface activation. XPS analysis suggests that the increase in functional binding sites (C=O) on the implant surface influences protein and cell adhesion [79].

## 3. Conclusions

In tissue engineering, there has long been a high level of interest in developing appropriate substitute materials that control cell behavior and thus regenerate the natural architecture and function of blood vessels. The basis of this is the development of a scaffold, either alone or in combination with drugs, growth factors, or other bioactive chemicals, that is capable of regulating cell behavior and thus the composition of the ECM. In addition, the mechanisms of neotissue formation and physiological healing must be considered when developing scaffolds for tissue engineering. For the development of new scaffolds, it is therefore not only necessary to use a biocompatible, resorbable, and hemocompatible material, but also to have a certain geometric structure so that there is an improvement in healing of the material used and thus restoration of tissue function. Surface modifications of biomaterials include not only targeted surface structuring, but also coating of the materials with bioactive substances. To increase the biocompatibility of biomaterials, their surface structure can be modified or adapted to the structure and properties of the natural ECM. Therefore, the efficient integration of a biomaterial and bioactive coating with endogenous degradable matrix proteins or surface geometry changes represents promising approaches. The results presented in this review show that the induction of angiogenesis at the implant site as well as the vascularization of biomaterials can be influenced by specific surface modifications. The neovascularization of a biomaterial can be supported by the application of pro-angiogenic substances as well as by biomimetic surface coatings and physicochemical surface activations. Furthermore, it has been confirmed that the geometric properties of the three-dimensional biomaterial matrix play a central role, as they guide or even enable the growth of blood vessels into a biomaterial. Optimization of biomaterial neovascularization could contribute to the development of tissue substitutes with higher mass and complexity, as well as to the development of novel and fast therapeutic modalities in reconstructive surgery.

## Figures and Tables

**Figure 1 bioengineering-09-00298-f001:**
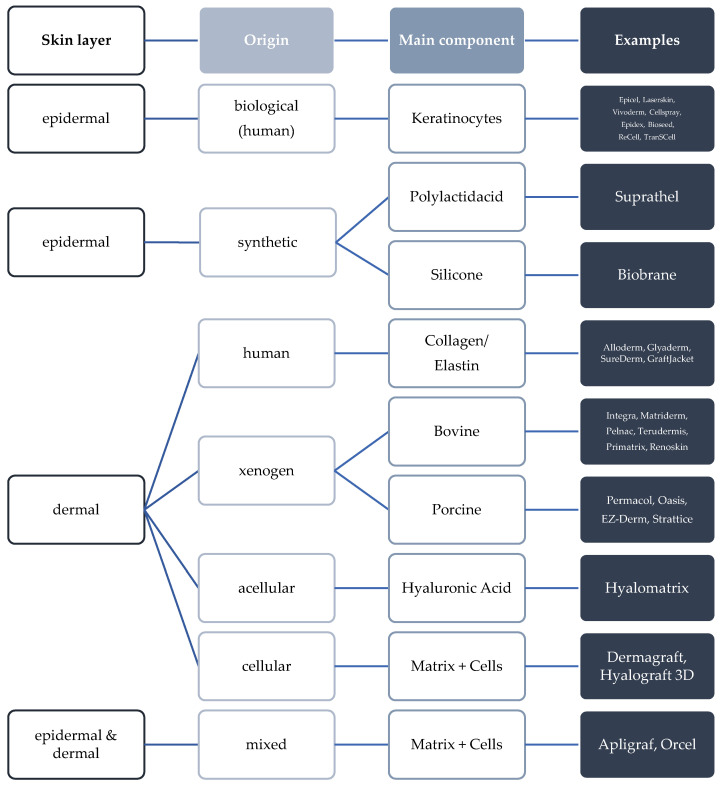
Overview of currently used skin substitute materials [7]. Depending on which skin layer is to be restored, a distinction is made between epidermal, dermal, and combined skin substitutes. The requirements for a dermal skin substitute are more challenging than for an epidermal skin substitute, which is mainly concerned with the rapid restoration of the epidermal structures. A dermal replacement should meet the best possible natural requirements of the body’s own dermis. This includes the reconstruction of the anatomy as well as the associated physiological function. Replacement materials are designed with their basic structure to control scarring, contraction, and pain in a positive sense by reducing healing time. All the sample materials mentioned are ^®^.

**Figure 2 bioengineering-09-00298-f002:**
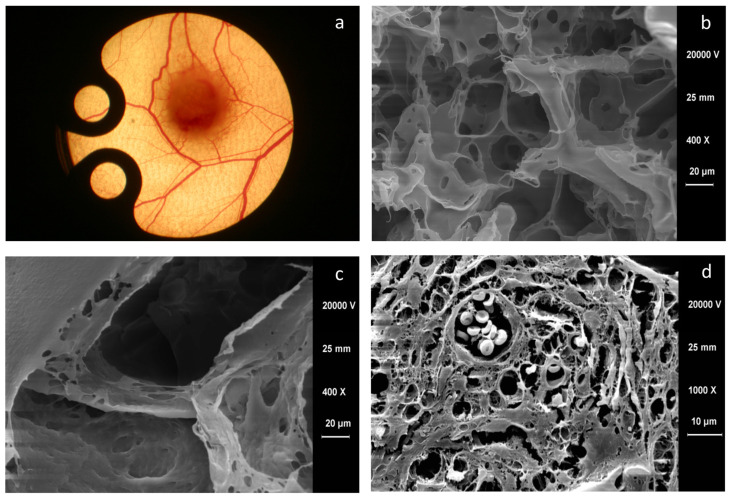
(**a**): Macroscopic image of a mouse dorsal skin fold chamber with implanted biomaterial. Of note, around the implanted material, a fine network of newly formed blood vessels can be seen. (**b**,**c**): Scanning electron micrographs of a highly porous, biodegradable 3-D matrix of lactocapromer terpolymer based on polylactide and polycaprolactone (**b**) and a biodegradable synthetic polyethylene glycol terephthalate/polybutylene terephthalate (PEGT/PBT)-block copolymer matrix with different porosity, 400× magnification. (**d**): Scanning electron micrographs of lactocapromer terpolymer with filled pores with multiple blood vessels, 1000× magnification.

**Table 1 bioengineering-09-00298-t001:** Summary of bone graft substitutes [8,9]. Bone graft substitutes are differentiated terminologically according to their origin. Composites of the individual substitute materials are not listed.

Autologous	Xenogen	Allogen	Alloplastic		Phycogenic
			*biological*	*synthetic*	Hydroxyapatite from 100% inorganic calcium phosphate, of which 95% is present as apatite; source material is calcareous encrusting marine algae
spongious	Bovine	Living donor	Hydroxyapatite	Ca_3_(PO_4_)_2_ cements	
cortico-spongious	Porcine	Cadaver donor	Platelet rich plasma	Hydroxyapatite	
vascular	Equine		CaSo_4_	β-tricalciumphosphate	
			Corals	Bioactive glasses	
				Polymer-based substitute materials	
				Metals	

**Table 2 bioengineering-09-00298-t002:** Material properties of the different used bone graft substitutes modified from [78].

	Material	Component	Ca/P Index	Porosity (%)
Cement	Calcibon ^®^	α-TCP, CaHPO4, CaCO_3_, pHA	1.57	8
	Biobon ^®^	α-TCP, DCPD	1.45	50–60
	Norian SRS ^®^	α-TCP, CaCO_3_, MCPM	1.67	
Granules	Algipore ^®^	Coraline HA	1.8–2.15	75–80
	BioOss ^®^	Bovine HA	2.03	59.7
	ChronOs ^®^	β-TCP	1.5	60
	Endobon ^®^	Bovine HA	1.67	45–85

*HA*: hydroxyapatite, *pHA*: precipitated hydroxyapatite, *TCP*: tricalcium phosphate, *DCPD*: dicalcium phosphate dihydrate, and *MCPM*: monocalcium phosphate monohydrate.

## Data Availability

The data sets used and analyzed during the current review are available from the respective corresponding authors of the cited published articles on reasonable request. No original or previously unpublished data were used in this review article.

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
