# Peer review of "Improving Vascularization of Biomaterials for Skin and Bone Regeneration by Surface Modification: A Narrative Review on Experimental Research"

_bioengineering, 2022, doi:10.3390/bioengineering9070298_

Round 1
Reviewer 1 Report
The manuscript describes the different fabrication methods for the enhancement of the neovascularization of injured skin and bone tissues. The authors shed light on the surface modified biomaterials that is engineered either using chemical and geometric modification of the surface. The authors include research works that utilized dorsal skinfold chamber model, which is a non-invasive microcirculatory analysis of striated muscle and skin tissue. Overall, the manuscript lacks detailed data and it needs improved organization.
Major issues:
- The manuscript covers very few research works in each topic. Should include more examples/studies from the literature.
- For instance, only single study using cathelicidin LL 37 was reported for angiogenesis under bioactive modification.
· The review largely highlights studies on skin tissue regeneration. Need to add examples and more details on bone tissue regeneration.
2. Should add more sections between introduction and conclusion. Within each subsection, applications can be categorized according to the tissue types such as skin, bone etc.
3. In section 1.5, the core characteristics of surface geometry that are required for vascularization should be incorporated.
- The authors stated that biocompatibility of the scaffold in the recipient tissue is a major challenge. Authors should address how surface modification can overcome these issues in the conclusion or discussion.
- A table summarizing the previous work done on improving the vascularization of biomaterials would enhance quality of the manuscript.
Minor concerns
1. Grammatical correction necessary in page 2, line 76.
- Need to make those complex sentences in the section 1.1, 1.2, and 1.4 into shorter sentences.
Author Response
First of all, we kindly thank reviewer#1 for critically reviewing our manuscript and her/his comments and would like to respond as follows:
Major issues:
- The manuscript covers very few research works in each topic. Should include more examples/studies from the literature. For instance, only single study using cathelicidin LL 37 was reported for angiogenesis under bioactive modification.
We have now added a lot more studies in each subsection and made more subsections between the bioactive modification options each for skin and bone, respectively. We therefore now changed the manuscript title and deleted “using the dorsal skin fold chamber” as many other studies have been integrated in the revised version not using this experimental setup. The new title reads as follows:” Improving vascularization of biomaterials for skin and bone regeneration by surface modification: a narrative review on experimental research”.
Please see the revised version of the manuscript marked by red font and underlining.
- The review largely highlights studies on skin tissue regeneration. Need to add examples and more details on bone tissue regeneration.
We thank the reviewer for this recommendation. According to the response no. 1, this has been added. Please see the revised version of the manuscript marked by red font and underlining.
- Should add more sections between introduction and conclusion. Within each subsection, applications can be categorized according to the tissue types such as skin, bone etc.
Please see responses to questions no 1-2. This recommendation has been fully implemented.
- In section 1.5, the core characteristics of surface geometry that are required for vascularization should be incorporated.
We thank the reviewer for this advice. Section 1.5. is now 2.7. in the revised version of the manuscript.
We have added the core characteristics, which now reads as follows: “In the circulatory system, blood vessels represent a unique anatomical structure, with their innate physiology forming the basis for mechanical support against blood pressure, vasoactive response, and cell functionality. Recently, biomaterials engineering has explored the design of anisotropic surface geometries, which analyzed direct biological effects on the replication of vascular tissue architecture. Anisotropy here refers to the ability of certain cells to exhibit different growth directions under the same conditions. A vascular scaffold therefore requires to be a biomaterial which is biodegradable/resorbable, biocompatible and hemocompatible. It must also display with robust mechanical properties and a large porosity enough for cell infiltration, cell-cell communication, and mass transportation.” marked by red font and underlining.
- The authors stated that biocompatibility of the scaffold in the recipient tissue is a major challenge. Authors should address how surface modification can overcome these issues in the conclusion or discussion.
We have now added other sentences to the conclusion section, which we hope clarifies this question. This reads now as follows: “In tissue engineering, there has long been a high level of interest in developing appropriate substitute materials that control cell behavior and thus regenerate the natural architecture and function of blood vessels. The basis of this is the development of a scaffold, either alone or in combination with drugs, growth factors or other bioactive chemicals, that is capable of regulating cell behavior and thus the composition of the ECM. In addition, the mechanisms of neotissue formation and physiological healing must be considered when developing scaffolds for tissue engineering. For the development of new scaffolds, it is therefore not only necessary to use a biocompatible, resorbable and hemocompatible material, but also to have a certain geometric structure, so that there is an improvement in healing of the material used and thus restoration of tissue function.” marked by red font and underlining.
- A table summarizing the previous work done on improving the vascularization of biomaterials would enhance quality of the manuscript.
We thank the reviewer for this recommendation and have added the new Figure 1 and the new Table 1 in the introduction section. Furthermore, the introduction section has been revised. Please see the revised version, marked by red font and underlining.
We have, however, summarized available skin and bone substitute materials in the figure and table, as we did not want to double data. Please understand, that we therefore have decided not to add another table for summarizing issues, which are described within this review article.
Minor concerns
- Grammatical correction necessary in page 2, line 76.
This has been corrected in the revised version of the manuscript.
- Need to make those complex sentences in the section 1.1, 1.2, and 1.4 into shorter sentences.
The sentences have been shortened and revised. However, after changes were made throughout the complete manuscript, attention was paid to this grammar problem and attempts were made to avoid it. Please see the respective parts in the revised version of the manuscript marked by red font and underlining.

Reviewer 2 Report
Organization of the manuscript is awkward. All the modification methods are currently under the introduction section. An independent section should be added such as Modification Methods. Then conclusions should be Section 3.
This review paper lacks the most updated technologies, designs and publications from other groups.
Sentences are wordy. Information density is low. For instance, Line 79-82 "The broad functional spectrum of LL 37 makes this effector molecule of the innate immune system, which combines antimicrobial, immunomodulatory, and proangiogenic properties, interesting for tissue engineering applications in the development of "functional" skin substitute tissue.” It is better to say it in a direct way: LL37 is a multi-functional molecule for tissue engineering that has antimicrobial, immunomodulatory, and proangiogenic properties. It is highly suggested that authors revise the manuscript again to make the statements more concise and informative.
Line 156. In the statement that "The PEGT/PBT-copolymer with different structural properties was used as a control." , “different structural properties” is too vague. More specific description should be given. In addition, PEGT/PBT-copolymer and lactocapromer terpolymer have degradation behavior. Is the observed enhanced vascularization due the degradation variance or surface properties?
The last paragraph of section 1.5 (Line 214 - 236) is more about chemical modification of the surface by plasma, not the geometrical tuning. It should be moved to somewhere else or as an independent section.
Author Response
Reviewer #2
First of all, we kindly thank reviewer#2 for critically reviewing our manuscript and her/his comments and would like to respond as follows:
- Organization of the manuscript is awkward. All the modification methods are currently under the introduction section. An independent section should be added such as Modification Methods. Then conclusions should be Section 3.
The sections have been corrected and newly ordered to gain more organization. Please see the revised version of the manuscript.
- This review paper lacks the most updated technologies, designs and publications from other groups.
We have now added a lot more studies in each subsection and made more subsections between the bioactive modification options each for skin and bone, respectively. We therefore now changed the manuscript title and deleted “using the dorsal skin fold chamber” as many other studies have been integrated in the revised version not using this experimental setup. The new title is as follows: ”Improving vascularization of biomaterials for skin and bone regeneration by surface modification: a narrative review on experimental research”.
Please see the revised version of the manuscript marked by red font and underlining.
- Sentences are wordy. Information density is low. For instance, Line 79-82 "The broad functional spectrum of LL 37 makes this effector molecule of the innate immune system, which combines antimicrobial, immunomodulatory, and proangiogenic properties, interesting for tissue engineering applications in the development of "functional" skin substitute tissue.” It is better to say it in a direct way: LL37 is a multi-functional molecule for tissue engineering that has antimicrobial, immunomodulatory, and proangiogenic properties. It is highly suggested that authors revise the manuscript again to make the statements more concise and informative.
The complete manuscript has been revised and corrected according to these suggestions.
- Line 156. In the statement that "The PEGT/PBT-copolymer with different structural properties was used as a control." , “different structural properties” is too vague. More specific description should be given. In addition, PEGT/PBT-copolymer and lactocapromer terpolymer have degradation behavior. Is the observed enhanced vascularization due the degradation variance or surface properties?
This sentence has been revised and reads now as follows: “The PEGT/PBT-copolymer was used as a control and displayed pore sizes of 250-300 µm, provided appropriate hydrophilicity to the copolymer and allowed high molecular permeability [9].” Marked by red font and underlining.
The reiver is right in his statement. We will not exclude that degradation also played a role in this experiment. We think that the vascularization is not due to the degradation process, which usually leads to a different signal communication of cytokines, which might be more anti-angiogenic. Furthermore, the degradation process takes way longer than the experiments took place in the described and cited studies. We therefore underscore that the seen processes were due to surface properties and their modifications.
- The last paragraph of section 1.5 (Line 214 - 236) is more about chemical modification of the surface by plasma, not the geometrical tuning. It should be moved to somewhere else or as an independent section.
Plasma and geometry were two different sections in our manuscript. For better understanding, we have reordered all sections and subsections and renamed them to reach more organization and structure.

Reviewer 3 Report
The authors made an interesting work summarizing on experimental research using the dorsal skinfold chamber in order to highlight the improving vascularization of biomaterials for skin and bone regeneration by surface modification: All this article is written referring to previous work and it does not make very much comparison with other works, from other authors, in order to emphasize some differences or similarities. So I understand that this article is not a review type article but an original article? The data presented here were reported in another article? It is not very clear…It is written in a very ambiguous and misleading way.
Minor modification:
- In the ABSTRACT, in order to avoid the use of two “and” from line 25 it should be written “physical or chemical surface”.
- In line 98, what does the expression: “Our intravital observations” represents?
Major modification:
1. Regarding the subsection 1.1 Challenges in skin and bone tissue regeneration with synthetic biopolymers and 1.2 . Bioactive modification of a biopolymer, I consider that it should be enlarged and more elaborated in discussions.
2. All the article to be re-written and restructured in a vigorous way that clarify if is a review article or an original article, where personal studies are analyzed and described.
Author Response
Reviewer #3
First of all, we kindly thank reviewer#3for critically reviewing our manuscript and her/his comments and would like to respond as follows:
- The authors made an interesting work summarizing on experimental research using the dorsal skinfold chamber in order to highlight the improving vascularization of biomaterials for skin and bone regeneration by surface modification: All this article is written referring to previous work and it does not make very much comparison with other works, from other authors, in order to emphasize some differences or similarities. So I understand that this article is not a review type article but an original article? The data presented here were reported in another article? It is not very clear…It is written in a very ambiguous and misleading way.
We thank the reviewer for this annotation. The submitted article was and is thought to be a review article on published work of mainly our own group. With the revision, this has significantly changed, as there have been added a lot of more other results as well. This underscores the form as review article. Furthermore, there was the need to change the article title to: “Improving vascularization of biomaterials for skin and bone regeneration by surface modification: a narrative review on experimental research”.
Minor modification:
- In the ABSTRACT, in order to avoid the use of two “and” from line 25 it should be written “physical or chemical surface”.
This has been addressed according to the reviewers recommendation. Please see the revised version of the manuscript marked by red font and underlining
- In line 98, what does the expression: “Our intravital observations” represents?
We have corrected this. This sentence now reads as follows: “The intravital microscopic observations showed that the intensity of new blood vessel formation in the peripheral zone of the implants as well as the speed and intensity of the extent of microvascular penetration in the center of the porous matrices differed significantly depending on the surface coating.” marked by red font and underlining in the revised version of the manuscript.
Major modification:
- Regarding the subsection 1.1 Challenges in skin and bone tissue regeneration with synthetic biopolymers and 1.2 . Bioactive modification of a biopolymer, I consider that it should be enlarged and more elaborated in discussions.
Subsections completely changed and enlarged due to other reviewer recommendations. The two sub sections are now sections 2.1. and 2.2.-3. Please see the revised version of these subsections.
- All the article to be re-written and restructured in a vigorous way that clarify if is a review article or an original article, where personal studies are analyzed and described.
We have now added a lot more studies in each subsection and made more subsections between the bioactive modification options each for skin and bone, respectively.
Please see the revised version of the manuscript marked by red font and underlining.

Round 2
Reviewer 1 Report
The response to the reviewers is missing, but it appears that all issues are well addressed.
Reviewer 2 Report
All comments were addressed. Check the spell and reference. Some mismatches detected.